# Semi-Supervised Anomaly Detection through Denoising-Aware Contrastive Distance Learning

## Abstract

Semi-supervised anomaly detection (AD) has garnered growing attention due to its ability to effectively combine limited labeled data with abundant unlabeled data. However, current methods often impose artificial constraints on the proportion of unlabeled anomalies in the training set or overlook potential noise from these anomalies, thereby impeding the effective training of models for anomaly detection in real-world scenarios where several anomalies may be present in the unlabeled dataset. Additionally, existing methods often struggle to effectively exploit and model the complex relationships between data instances, which is critical for learning more discriminative features and accurate distance measures. Distance-based methods, in particular, typically rely on Euclidean distance metric, which lacks the flexibility to capture complex correlations across different data dimensions. To address above challenges, we propose CAD, a denoising-aware Contrastive distance learning framework for semi-supervised AD. It introduces a contrastive training objective to facilitate the learning of distinctive representations by contrasting the average distance between anomalies and unlabeled samples. To fully exploit the information from the unlabeled data meanwhile mitigate the effects of noise, we incorporate a two-stage anomaly denoising and expansion strategy to refine the dataset by identifying high-confidence samples from the unlabeled set. Furthermore, we employ a parameterized bilinear tensor distance layer to learn a customized distance metric, enabling the model to capture intricate relationships among data points. Extensive experiments on 10 real-world datasets demonstrate that CAD significantly outperforms existing semi-supervised AD models. Code available at https://github.com/CADrepo/CAD.

**ACM Reference Format:**
Anonymous Author(s). 2024. Semi-Supervised Anomaly Detection through Denoising-Aware Contrastive Distance Learning. In *Proceedings of The Web Conference 2025 (WWW'25)*. ACM, New York, NY, USA, 9 pages. https://doi.org/10.1145/nnnnnnn.nnnnnnn

## 1 Introduction

Anomaly detection refers to identifying data points that deviate markedly from other samples [11]. This critical area of research in data mining and information retrieval has wide-ranging applications across various domains, including disease diagnosis [9, 15], fraud detection [7, 22], network intrusion detection [10], data

preprocessing for machine learning [32]. While obtaining labels for anomalies can be challenging, real-world scenarios often provide access to a limited amount of labeled data (e.g., clinically confirmed cases, authenticated security breaches). Consequently, semi-supervised anomaly detection has garnered growing interest in recent years. This approach leverages the limited labeled data, combining the strengths of supervised and unsupervised methods to improve detection accuracy and generalization.

With the rapid advancement of deep learning, recent AD models have increasingly adopted neural networks to enhance their ability to capture complex data patterns. These models are designed with different approaches and objectives, shaped by varying perspectives on the nature of anomalies. For instance, regression-based methods learn an end-to-end scoring function to distinguish between normal and anomalous instances [27, 41], which consider individual data points without accounting for relationships between points. Reconstruction-based methods employ models such as autoencoders [24, 42] or GANs [2, 37] to identify anomalies by comparing data points with their reconstructed versions. Distance-based methods compute distances between data points, considering those far from others as anomalies [25, 28, 35]. Among these approaches, distance-based methods inherently incorporate the relationships between data points for anomaly identification. Recent advancements in neural representation learning have significantly propelled the development of distance-based methods. This paper primarily focuses on distance-based anomaly detection techniques.

Despite the progress made by current distance-based AD approaches, several significant challenges still remain unsolved. Firstly, while existing methods consider relationships between normal and anomalous instances, they often underutilize the rich information and the complex interrelations among data instances. For example, these methods typically focus on comparing pairs of data points with rigid loss functions like hinge loss, which can result in suboptimal representations[1]. Secondly, current approaches typically train AD models on basically clean datasets, which means almost all of the unlabeled data are normal instance. In practice, researchers often artificially control the proportion of unlabeled anomalies in the training set to be very low (e.g., less than 2% [26, 27, 42]). This approach, however, may not reflect real-world scenarios where the unlabeled data can contain many anomalies. Existing approaches often overlook the noise in unlabeled data, which may prevent them from learning robust data representations for anomaly detection. Moreover, existing distance-based AD models often employ conventional distance measures like Euclidean distance, which lacks the flexibility needed to effectively capture complex correlations across different data dimensions in real-world datasets.

To tackle those issues, we propose CAD, a denoising-aware Contrastive distance learning framework for semi-supervised AD

---

[1]The optimization process will cease once the margin is satisfied, leading to insufficient separation of hard-to-detect anomalies.

which effectively leverages the inherent relationships within data to identify subtle anomalies. The framework employs a contrastive distance learning objective to learn discriminative representations, fully exploiting the abundance of relational information in the data. To further enhance the utility of unlabeled data and mitigate noise, CAD incorporates a two-stage anomaly denoising and expansion strategy, which introduces additional high-confidence samples into the training process. Furthermore, instead of relying on Euclidean distance, CAD utilizes a parameterized bilinear tensor distance to capture the complex feature correlations. The anomaly scores are computed based on deviations from the global context. We conduct experiments on 10 commonly used real-world AD datasets that are commonly used by existing approaches to evaluate CAD. Experimental results show that CAD significantly outperforms existing semi-supervised AD models. Additional ablation studies also demonstrate the effectiveness of individual components of our model. In summary, our contributions are as follows:

- We propose a denoising-aware contrastive distance learning framework that contrasts the average distances between anomalies and unlabeled samples, enabling the model to learn discriminative features for semi-supervised AD.
- We design a two-stage anomaly denoising and expansion strategy in contrastive training to maximize the utilization of unlabeled data while mitigate the impact of contamination in data.
- We introduce a parameterized bilinear tensor distance layer to learn a customized metric, enabling the model to capture intricate relationships and measure divergence among data instances.
- We conduct experiments on 10 commonly used real-world AD datasets, which show that CAD greatly outperform existing semi-supervised AD models.

## 2 Related Works

### 2.1 Unsupervised Anomaly Detection

The earlier works often focus on anomaly detection in an unsupervised manner due to the difficulty of accessing labeled data. Different methods define anomalies from various perspectives, leading to a diverse range of unsupervised anomaly detection techniques. One of the widely used methods is the Local Outlier Factor (LOF) [4], which compares the local density of a point to that of its neighbors. Another classical approach is Isolation Forest [20, 21], which isolates anomalies by randomly partitioning the feature space and identifying points that are easily separated from the rest. One-class classification methods, such as One-Class SVM [31] and Deep SVDD [28], aim to learn a boundary that encloses normal data points, treating outliers as anomalies. Distribution-based methods estimate the distribution of the data and consider the data points at the tail of the distribution as anomalies, such as ECOD [19] and COPOD [18]. Recent unsupervised techniques leverage the power of deep learning. Autoencoders [5, 17, 40] learns compact representations of normal data, with anomalies identified as points with high reconstruction error. Similarly, generative adversarial networks [23, 30, 36] learns to generate normal samples and using the discriminator to identify anomalies. To address the limitations of relying on a single method, ensemble-based approaches such as LSCP [38] and MetaOD [39] have been developed, which aim to

automatically select or combine multiple detectors to improve robustness and accuracy. Despite these advancements, unsupervised models still suffer from low model performance due to the lack of prior knowledge about the intrinsic characteristics of anomalies. This limitation highlights the need for more adaptive and context-aware approaches that can better capture the complex and varied nature of anomalies in real-world datasets.

### 2.2 Semi-supervised Anomaly Detection

In recent years, semi-supervised anomaly detection has emerged to address real-world scenarios where limited labeled samples are available. Effective semi-supervised anomaly detection models typically need to combine elements of both supervised and unsupervised learning paradigms to improve detection accuracy and generalization. Several semi-supervised approaches incorporate supervised signals into unsupervised models to guide the training process. For example, REPEN [25] enlarges the distance between the representation of normal and anomalous samples to learn discriminative representations for anomaly detection. GANormaly [2] uses the labeled data to guide the generator to produce more realistic normal samples in GAN, enhancing the model's ability to distinguish anomalies. DeepSAD [29] extends the unsupervised Deep SVDD approach by incorporating labeled data to guide the learning process, aiming to map normal instances close to a hypersphere's center and anomalies far from it. FEAWAD [40] utilizes label information to guide the training of autoencoders, improving their capacity to extract features relevant to anomaly detection. The other approaches directly learn the anomaly scores to obtain data representations associated with anomaly detection. For example, DevNet [27] learns a neural network to map data instances into scalar anomaly scores, guided by a reference distribution of anomaly scores. PReNet [26] employs a relation network to predict pairwise relationships for anomaly detection, which can also be interpreted as a special distance metric.

However, current semi-supervised approaches often overlook the potential noise in the training data, which limits the model's ability to learn distinctive data representations necessary for effective anomaly detection. In addition, existing distance-based methods typically employ conventional distance functions without learnable parameters, which can be insufficiently flexible to capture complex correlations across dimensions. To address this limitation, we propose a two-stage denoising-aware contrastive training framework with a parameterized bilinear distance metric to fully leverage and model the abundant relationships between data, enabling the learning of distinctive representations.

## 3 Methodology

### 3.1 Problem Statement

Given a dataset $\mathcal{X} = \{x_1, x_2, ..., x_n, x_{n+1}, x_{n+2}, ..., x_{n+m}\}$, where $x_i \in \mathbb{R}^{d_i}$ is a $d_i$-dimension data point. In semi-supervised AD, we have a limited number $m$ of labeled anomalies $\mathcal{X}_A = \{x_{n+1}, x_{n+2}, ..., x_{n+m}\}$, and a large pool of unlabeled data $\mathcal{X}_U = \{x_1, x_2, ..., x_n\}$, where $m << n$. An anomaly detection model aims to learn a scoring function $\phi : \mathcal{X} \rightarrow \mathbb{R}$ that assigns an anomaly score to data instances such that for any anomalous sample $x_i$ and normal sample $x_j$, the inequality holds: $\phi(x_i) > \phi(x_j)$. Considering that the unlabeled data

compromise the majority of the data and may contain anomalies, how to effectively leverage the rich information from unlabeled data is an important problem in anomaly detection.

## 3.2 Model Overview

The overall architecture of CAD is shown in Figure 1. CAD learns the scoring function by contrasting the distance between data samples. To enable the learning of distinctive data representations along with network parameters, we propose a novel contrastive learning-based objective that fully exploits the relationships among data samples. This objective encourages larger distances between anomalies and unlabeled instances, while promoting closer distances among similar unlabeled instances. To effectively leverage unlabeled data and mitigate the influence of noise, CAD employs a two-stage anomaly denoising and expansion strategy. This approach provides high-confidence training samples by excluding noise from training data and dynamically identifying anomalies for training. Additionally, to address the limitations of conventional distance metrics in capturing complex inter-dimensional correlations, we introduce a parameterized distance model that learns a context-specific distance metric tailored for anomaly detection. Finally, the anomaly score of a data instance is determined by its deviation in distance from the global context.

## 3.3 Denoising-aware Contrastive Learning

While existing methods consider relationships between normal and anomalous instances, they typically focus on comparing limited pairs of data points, often using rigid loss functions like hinge loss. These methods enforce a fixed margin between anomalies and normal instances, which can result in suboptimal representations. Recently, contrastive learning has proved to be an effective technique in representation learning [6, 13]. By leveraging the relationships between positive and a pool of negatives, contrastive learning utilizes a soft exponential loss to learn discriminative representations. In this paper, we consider leveraging contrastive learning for anomaly detection, aiming to enhance the discriminative power of learned representations through this flexible and effective framework. In the presence of supervised signals, this objective can be extended to enlarge the similarity between samples from the same classes while pushing instances from different classes farther apart [16], which is implemented using the following InfoNCE loss:

$$\ell(x, x_+, \mathcal{X}_-) = -log \frac{e^{(\mathcal{S}(f(x),f(x_+)))}}{e^{(\mathcal{S}(f(x),f(x_+)))} + \sum_{x_- \in \mathcal{X}_-} e^{(\mathcal{S}(f(x),f(x_-)))}}$$

where $x$, $x_+$ and $\mathcal{X}_-$ are often denoted as anchor, positive sample and the negative sample set. The function $\mathcal{S}$ denotes the similarity/distance metric (e.g., cosine similarity), $f : \mathbb{R}^{d_i} \to \mathbb{R}^{d_h} (d_h < d_i)$ is the underlying representation learning function that maps the sample into a lower-dimensional representation space with more distinguishable features.

However, in the context of multi-dimensional data anomaly detection, relying on the similarity of normal-anomalous pairs as negative samples can be limiting. This is because the number of available anomalies is often insufficient to effectively drive contrastive learning in semi-supervised AD. Additionally, the presence of noise in unlabeled datasets can lead to incorrect associations

between samples, further degrading the quality of the representations learned. To tackle these challenges, we propose a two-stage denoising-aware contrastive learning approach. This approach incorporates an *anomaly denoising* and *expansion* strategy to reduce the impact of noise and enrich the training data with valuable anomaly samples. Our method enhances contrastive learning by adjusting the training objective to maximize the distance between anomalies and the normal set, while utilizing the abundant normal-normal pairs as informative negative samples. This strategy leverages the numerous relationships within the data, allowing the model to learn more distinctive and robust representations. Consequently, our approach improves anomaly detection performance by effectively addressing noise and scarcity of anomalies in the training data.

*3.3.1 Stage 1 - Normal Sample Denoising.* In the first stage, the goal is to refine the unlabeled dataset by excluding instances that are likely anomalous, thereby retaining predominantly normal samples for model training. This is achieved through an unsupervised anomaly detection method, which assigns a rough anomaly score $\phi_r(x_i)$ to each instance in the unlabeled set. In our implementation, we employ a distance-based method CBLOF [14] for initial anomaly scoring[2]. After scoring, we exclude instances likely to be anomalous by using the mean value $\mu$ and the standard deviation $\sigma$ of the anomaly scores:

$$\mathcal{X}_{\widehat{U}} = \{x_i | \phi_r(x_i) < \mu + \alpha\sigma\} \tag{1}$$

where $\alpha$ is a hyperparameter that controls the strictness of the threshold. Consistent with REPEN [25], an anomaly threshold $\mu + \alpha\sigma$ results in a false positive upper bound of $\frac{1}{1+\alpha^2}$. A larger $\alpha$ selects fewer pseudo anomalies, which is suitable for cleaner datasets, whereas a smaller $\alpha$ results in a smaller $\mathcal{X}_{\widehat{U}}$, appropriate for more contaminated datasets. We will thoroughly investigate the impact of varying $\alpha$ in Section 4.5.

Since the predictions of unsupervised method may not be accurate, the anomaly set predicted by unsupervised method $\mathcal{X}_U / \mathcal{X}_{\widehat{U}}$ will not be used directly. Given that the refined unlabeled set $\mathcal{X}_{\widehat{U}}$ predominantly consists of normal samples, they should exhibit higher similarity to each other compared to anomalies. To enforce this, our proposed loss function maximizes the distance between anomalies and the refined unlabeled set while minimizing the distance between normal samples. To train the model, we first calculate the distance between a data sample and a set. Specifically, given a distance function $\mathcal{S}$ that measures the distance between two data points, the distance between a sample $x$ and the refined unlabeled set $\mathcal{B}_{\widehat{U}}$ is defined as the average distance:

$$\mathcal{D}(x, \mathcal{B}_{\widehat{U}}) = \frac{1}{|\mathcal{B}_{\widehat{U}}|} \sum_{x_u \in \mathcal{B}_{\widehat{U}}, x_u \neq x} \mathcal{S}(x, x_u) \tag{2}$$

We then train the model by contrasting the distance between data points. Specially, let $\mathcal{B}_A \in \mathcal{X}_A, \mathcal{B}_{\widehat{U}} \in \mathcal{X}_{\widehat{U}}$ be anomalous set and refined unlabeled set in a batch, the loss function is defined as:

$$\mathcal{L}_1 = \frac{1}{|\mathcal{B}_A|} \left( \sum_{x_a \in \mathcal{B}_A} - \log \frac{e^{\mathcal{D}(x_a, \mathcal{B}_{\widehat{U}})}}{e^{\mathcal{D}(x_a, \mathcal{B}_{\widehat{U}})} + \sum_{x_u \in \mathcal{B}_{\widehat{U}}} e^{\mathcal{D}(x_u, \mathcal{B}_{\widehat{U}})}} \right) \tag{3}$$

---

[2]A variety of established unsupervised models can be employed here (e.g., Sp [34], IForest [20]), yielding comparable performance for CAD in our preliminary experiments.

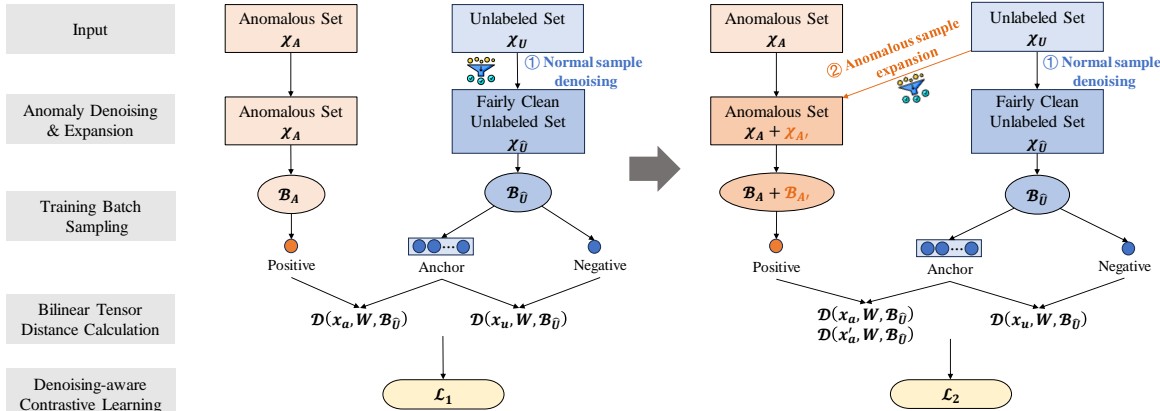

**Figure 1: Overall architecture of proposed CAD.**

In this formulation, the entire set $\mathcal{B}_{\widehat{U}}$ serves as the anchor, with $x_a$ and $x_u$ acting as positive and negative samples, respectively. This formulation presents two key benefits. First, unlike regression-based losses that focus on individual data points, this contrastive loss exploits the relative distances between numerous data pairs, capturing richer and more informative relationships. As demonstrated by [16], contrastive loss not only leads to improved classification accuracy but also enhances model robustness compared to traditional classification losses. Second, rather than using a single normal sample as the anchor and calculating the loss for each instance in $\mathcal{B}_{\widehat{U}}$, using the entire set $\mathcal{B}_{\widehat{U}}$ as the anchor and computing the average distance reduces the computational complexity while maintaining comparable model performance.

*3.3.2 Stage 2 - Anomalous Sample Expansion.* In the second stage, we aim to expand the set of labeled anomalies by dynamically identify anomalies from $\mathcal{X}_U$. Labeled anomalies provide valuable insights into the nature of anomalous instances during training, but their scarcity limits the effectiveness of the model. To tackle this, after Stage 1 training is complete, we select the samples with the $k_A$ highest anomaly scores as expanded anomalies at the end of each training epoch, where $k_A = |\mathcal{X}_U/\mathcal{X}_{\widehat{U}}|$ is the number of instances excluded in the first stage. This forms a pseudo anomaly set $\mathcal{X}_{A'}$:

$$\mathcal{X}_{A'} = \left\{ x_i \in \mathcal{X} \mid i \in \arg\max_j \phi(x_j),\ j \in \{1, \ldots, k_A\} \right\} \quad (4)$$

Next, we incorporate the pseudo-labeled anomalies into the training process. Specifically, let $\mathcal{B}_{A'} \in \mathcal{X}_{A'}$ be the set of pseudo anomalies in a batch. The training objective is then revised as follows:

$$\mathcal{L}_2 = \frac{1}{n_A} \Big( \sum_{x_a \in \mathcal{B}_A} - \log \frac{e^{\mathcal{D}(x_a, \mathcal{B}_{\widehat{U}})}}{e^{\mathcal{D}(x_a, \mathcal{B}_{\widehat{U}})} + \sum_{x_u \in \mathcal{B}_{\widehat{U}}} e^{\mathcal{D}(x_u, \mathcal{B}_{\widehat{U}})}}$$
$$- \lambda \sum_{x'_a \in \mathcal{B}_{A'}} \log \frac{e^{\mathcal{D}(x'_a, \mathcal{B}_{\widehat{U}})}}{e^{\mathcal{D}(x'_a, \mathcal{B}_{\widehat{U}})} + \sum_{x_u \in \mathcal{B}_{\widehat{U}}} e^{\mathcal{D}(x_u, \mathcal{B}_{\widehat{U}})}} \Big) \quad (5)$$

where $n_A = |\mathcal{B}_A| + |\mathcal{B}_{A'}|$ is the total number of anomalies in the batch. We introduce a loss weight $\lambda$ to control the impact of expanded anomalies. Intuitively, a larger $\lambda$ assigns higher weights, which is appropriate for high-quality expanded anomalies. The exploration of $\lambda$ will be discussed in Section 4.5.

### 3.4 Bilinear Tensor Distance

To address the limitations of traditional distance measure in capturing complex correlations across dimensions, we introduce the parameterized distance function to capture complex correlations between features. After obtaining the representations, we define a parameterized distance function in a bilinear tensor product manner. This approach allows us to model complex relationships between data points by introducing a learnable distance metric. Specifically, given the encoder $f$ parameterized by $\theta$, a distance metric tensor $W \in \mathbb{R}^{d_h \times d_h \times c}$ and a pair of data instances $x_1, x_2$, the distance function $\mathcal{S}$ in Eq (2) can be redefined as:

$$\mathcal{S}(x_1, W, x_2) = \frac{1}{c} \sum_{i=1}^{c} \tanh(f(x_1, \theta) W^i f(x_2, \theta)^T) \quad (6)$$

where $W^i$ refers to the $i_{th}$ slice in the third dimension of $W$. $c$ refers the number of channels in the distance function, which is associated with the representation size $c = \beta d_h$. The tanh non-linearity is applied to introduce flexibility and allow for more complex patterns to be captured in the distance computation. This distance function is inspired by the neural tensor network [33], which also models interactions between data points using tensor products. In our case, we average the $c$ dimensions to compute a single distance score. This can be interpreted as aggregating the distances computed from $c$ different perspectives, akin to a multi-channel approach, where each slice of the tensor captures a different aspect of the relationship between $x_1$ and $x_2$. Consequently, this operation is analogous to an average pooling mechanism and provides a more comprehensive assessment of the relationship between two data points.

*Overall Training.* Algorithm 1 present training procedure of CAD. An unsupervised AD model is first applied to perform normal sample denoising in line 1. And then, the model parameters are randomly initialized in line 2. After that, lines 3-16 describe the training process. In particular, line 5-8 randomly samples a batch of instances to train the model. Considering that the labeled instances are rare, the sampling process of labeled anomalies are independent from that of unlabeled data to ensure there are labeled anomalies in each batch. That means the labeled instances may be repeatedly used in different batches. Since the model may not be fully trained at the beginning, the anomaly sample expansion process will begin after

---

**Algorithm 1** CAD Training

---

**Input:** training set $\mathcal{X} = \{\mathcal{X}_U, \mathcal{X}_A\} \in \mathbb{R}^d$
**Output:** network parameters $\Theta = \{\theta, W\}$
1: normal label denoising to generate $\mathcal{X}_{\widehat{U}}$
2: randomly initialize $\Theta$
3: **for** $i = 1$ to $n\_epochs$ **do**
4:     **for** $j = 1$ to $n\_batches$ **do**
5:         sample training batch $\mathcal{B}_A, \mathcal{B}_{\widehat{U}}$ from $\mathcal{X}_A, \mathcal{X}_{\widehat{U}}$
6:         **if** $i > n\_epochs/2$ **do**
7:             sample training batch $\mathcal{B}_{A'}$ from $\mathcal{X}_{A'}$
8:         **end if**
9:         calculate $d(x, W, \mathcal{B}_{\widehat{U}})$ for each $x \in \mathcal{B}$
10:         calculate loss value using Eq (3), (5)
11:         gradient descent to optimize network parameters $\Theta$
12:     **end for**
13:     **if** $i \geq n\_epochs/2$ **do**
14:         anomalous label expansion to generate $\mathcal{X}_{A'}$
15:     **end if**
16: **end for**
17: **return** $\Theta$

---

$n\_epochs/2$ epochs of training. After calculating the loss value, we use an Adam optimizer to update model parameters.

## 3.5 Inference

After $f$ and $D$ are trained, we leverage the distance between data points to detect anomalies. Rather than calculating the anomaly score as the average distance between a target instance and all points in $\mathcal{X}_U$, we simplify the process by utilizing the global context $\bar{z}$. Specially, with the learned parameters $\Theta = \{\theta, W\}$, the anomaly score for an instance $x$ is calculated as follows:

$$\phi(x, \Theta) = \frac{1}{c} \sum_{i=1}^{c} (f(x, \theta) W^i \bar{z}^T) \tag{7}$$

This approach significantly reduces the computational complexity, as it requires only a single bilinear distance computation, rather than $|\mathcal{X}_U|$ computations. We use the average representation of $\mathcal{X}_U$ to approximate $\bar{z}$, since the majority of $\mathcal{X}_U$ are normal samples [3]:

$$\bar{z} = \frac{1}{|\mathcal{X}_U|} \sum_{x_i \in \mathcal{X}_U} f(x_i, \theta) \tag{8}$$

In practical applications, $\bar{z}$ can be precomputed and stored after the training phase. Based on our preliminary experiments, this approach demonstrates performance comparable to the more computationally intensive method.

## 4 Experiments

### 4.1 Experimental Setup

**Datasets.** We use 10 real-world datasets from various domains that are commonly used by related works [12, 19, 20, 26, 27, 42] to evaluate the effectiveness of different anomaly detection models. The datasets can be accessed from the ODDS library[4] and the

---

[3]We also evaluate the average embedding of $\mathcal{X}$ or $\mathcal{X}_{\widehat{U}}$ in our experiment, yielding comparable performance results.
[4]https://odds.cs.stonybrook.edu/

**Table 1: Statistics of datasets. N is the number of instances, $d_i$ is the dimension, $N_A$ is the number of anomalies, $f_A$ refers to the number of labeled anomalies and the ratio w.r.t. the number of all anomalies in the training set.**

| Datasets | N | $d_i$ | $N_A$ | $f_A$ | Category |
|---|---|---|---|---|---|
| Cardiotocography | 2114 | 21 | 466 | 18(5%) | Healthcare |
| Mammography | 11183 | 6 | 260 | 10(5%) | Healthcare |
| Musk | 3062 | 166 | 97 | 3(5%) | Chemistry |
| Waveform | 3443 | 21 | 100 | 4(5%) | Physics |
| SpamBase | 4207 | 57 | 1679 | 13(1%) | Document |
| Satellite | 6435 | 36 | 2036 | 16(1%) | Image |
| Mnist | 7603 | 100 | 700 | 28(5%) | Image |
| Campaign | 41118 | 62 | 4640 | 37(1%) | Finance |
| Fraud | 284807 | 29 | 492 | 19(5%) | Finance |
| Census | 299285 | 500 | 18568 | 148(1%) | Sociology |

Adbench benchmark [12]. We follow existing works [1, 3, 12, 20] to define anomalies for each dataset according to domain-specific knowledge or using the minority classes. In detail, *Cardiotocography* and *Mammography* are about disease diagnosis, and the data points with specific disease will be treated as anomalies. The *Musk* dataset is to predict new molecules to be musks or non-musks, and the musk classes with fewer samples are treated as anomalies. The *Waveform* dataset contains three classes of waves, and the first class is used as normal class and the rest two classes are sampled as anomalies. *Spambase* is a spam email detection task, and the anomalies are spam emails. *Satellite* and *Mnist* are image classification datasets from astronautics and hand-written letter recognition domains, and the classes with fewer samples will be treated as anomalies. *Campaign* is a bank telephone promotion dataset, where rarely successful records are treated as anomalies. *Fraud* is a credit card fraud detection task, and the anomalies are fraudulent records. *Census* dataset is from US census bureau dataset, and the goal is to find the rare persons with high income.

We randomly sample 80% data points for training and leave the rest 20% as testing data. Note that when partitioning the dataset into training and test sets, unlike some existing works that only sample a small fraction of unlabeled anomalies in the training data (commonly 2% of all anomalies [26, 27, 42]), the labels for normal and anomalous instances were stratified to maintain their original proportions, which means there may be more unlabeled anomalies in the training set. We randomly select a small set of anomalous instances in the training set as labeled data. The ratio of labeled data depends on the scale of datasets. Generally, for datasets with anomalies less than 1000, we randomly sample 5% of labeled anomalous instances, for datasets with anomalies more than 1000, the ratio is 1%. The details of the datasets are listed in Table 1. A z-score normalization is performed for all the datasets, and the dimensions with a standard deviation equal to 0 will be dropped in this stage, since all of the instances have the same value in those dimensions, which brings no additional information for identifying anomalies. The models with the lowest loss value in the training set will be used to evaluate on the test set.

**Evaluation Metrics.** Two widely adopted metrics are used to evaluate the effectiveness of anomaly detection models, *i.e.,* the Area Under Receiver Operating Characteristic Curve (AUC-ROC) and the Area Under Precision-Recall Curve (AUC-PR), where ROC

Table 2: Overall comparison on 10 real-world datasets. Methods with the best performance are marked in bold.

| Datasets | AUC-ROC | | | | | | | AUC-PR | | | | | | |
|---|---|---|---|---|---|---|---|---|---|---|---|---|---|---|
| | GAN | REPEN | DevNet | DeepSAD | FEAWAD | PReNet | CAD | GAN | REPEN | DevNet | DeepSAD | FEAWAD | PReNet | CAD |
| Cardiotocography | 0.7018 | 0.8234 | 0.8692 | 0.7832 | 0.7852 | 0.8441 | **0.9264** | 0.4315 | 0.6017 | 0.7361 | 0.5453 | 0.6672 | 0.7068 | **0.7929** |
| Mammography | 0.8568 | 0.8984 | 0.9025 | 0.8925 | 0.8799 | 0.9078 | **0.9092** | 0.1937 | 0.4563 | 0.5604 | 0.4698 | 0.5343 | 0.5511 | **0.5618** |
| Musk | **1.0000** | **1.0000** | 0.8027 | 0.9639 | 0.8580 | 0.9076 | **1.0000** | **1.0000** | **1.0000** | 0.7548 | 0.7925 | 0.7926 | 0.8932 | **1.0000** |
| Waveform | 0.6068 | 0.8046 | 0.7900 | 0.7042 | 0.7088 | 0.7853 | **0.8834** | 0.0637 | 0.0945 | 0.1947 | 0.1787 | 0.1761 | 0.1876 | **0.3420** |
| SpamBase | 0.5193 | 0.7159 | 0.6179 | 0.5566 | 0.6299 | 0.6141 | **0.7823** | 0.4096 | 0.5780 | 0.5948 | 0.4393 | 0.5914 | 0.6014 | **0.7297** |
| Mnist | 0.6877 | 0.9671 | 0.8801 | 0.9016 | 0.9042 | 0.8672 | **0.9676** | 0.2607 | 0.7848 | 0.7231 | 0.5980 | 0.6486 | 0.7036 | **0.8247** |
| Satellite | 0.7231 | 0.7571 | 0.8077 | 0.7963 | 0.7404 | 0.7707 | **0.8543** | 0.5507 | 0.7445 | 0.7298 | 0.6798 | 0.6854 | 0.7030 | **0.8069** |
| Campaign | 0.6607 | 0.7996 | 0.7432 | 0.7301 | 0.7682 | 0.7244 | **0.8718** | 0.2198 | 0.3977 | 0.3466 | 0.2751 | 0.3450 | 0.3300 | **0.4707** |
| Fraud | 0.7526 | 0.9683 | 0.9432 | 0.9430 | 0.9479 | 0.9435 | **0.9734** | 0.1333 | 0.6562 | 0.6116 | 0.5868 | 0.5727 | 0.5697 | **0.7423** |
| Census | 0.6904 | 0.8945 | 0.7991 | 0.7266 | 0.8020 | 0.7876 | **0.8949** | 0.1006 | 0.3701 | 0.3586 | 0.1860 | 0.2000 | 0.3548 | **0.4543** |
| Average | 0.7199 | 0.8629 | 0.8156 | 0.7988 | 0.8025 | 0.8152 | **0.9063** | 0.3364 | 0.5684 | 0.5611 | 0.4751 | 0.5213 | 0.5601 | **0.6725** |
| P-value | 0.0002 | 0.0118 | 0.0010 | 0.0013 | 0.0001 | 0.0002 | - | 0.0001 | 0.0015 | 0.0004 | 4.04e-6 | 2.34e-5 | 2.24e-5 | - |

curve plots the true positive rate against the false positive rate, while the PR curve plots precision against recall. Both of the metrics has the bound of [0,1], and a higher value means a better model performance. While both metrics provide valuable insights into model performance, they offer distinct perspectives on the classification task. AUC-ROC is insensitive to class imbalance and provides an overall measure of model accuracy across all possible thresholds. In contrast, AUC-PR is particularly sensitive to the performance on the minority class, making it especially relevant in highly imbalanced datasets typical of anomaly detection scenarios [8].

**Baselines.** We consider six state-of-the-art semi-supervised anomaly detection models GAN [2], REPEN [25], DevNet [27], Deep-SAD [29], FEAWAD [42] and PReNet [26] as our competing methods. To test the data efficiency of the semi-supervised AD models, a popular unsupervised AD model IForest [21] is also added for comparison. Among these methods, REPEN and DeepSAD apply a Euclidean distance function to identify anomalies. PReNet learns pairwise relationships among normal and anomalous instances, which can also be interpreted as a special distance metric.

**Parameter Settings.** In our implementation, a multilayer perceptron (MLP) with a single hidden layer is used to learn representations from data. Given a dataset with the dimension of $d_i$, the representation size of the output layer $d_h$ is set to $\min\left(32, \max\left(4, \frac{d_i}{4}\right)\right)$. Based on the experimental results in Section 4.5, the number of channels of bilinear tensor distance model $c$ is set to 3 times of the embedding size. The weight of the loss derived from pseudo-labels, $\lambda$, is set to 0.3, while the hyperparameter $\alpha$ controlling the anomaly threshold in the first training stage is set to 3. Model parameters are optimized using the Adam optimizer, with a weight decay of 0.2. Consisting with previous works [12], the models with the lowest loss value in the training set will be used to evaluate on the test set. For the baseline methods, we follow the existing works and use their released code with the default parameter settings.

## 4.2 Main Results

We conducted a comparative analysis of the proposed method against baseline approaches using 10 real-world datasets. Given the limited labeled data, different data divisions can lead to varying model performances. Therefore, we report the average results from ten independent experiments for all models. The results are presented in Table 2. As shown in the table, the proposed method outperforms other methods, achieving the highest AUC-ROC and AUC-PR scores across all 10 datasets. In terms of AUC-ROC, the proposed method, CAD, shows notable average improvements over REPEN (5.0%), DevNet (11.1%), PReNet (11.2%), FEAWAD (12.9%), DeepSAD (13.5%), and GAN (25.9%). For AUC-PR, CAD achieves even more substantial improvements compared to REPEN (18.3%), DevNet (19.9%), PReNet (20.0%), FEAWAD (29.0%), DeepSAD (41.5%), and GAN (99.9%). These results demonstrate the effectiveness of our proposed approach across various settings.

We also observe that CAD achieves significant improvements in some datasets (e.g., SpamBase, Campaign), while the improvements are relatively smaller in other datasets (e.g., Mammography). One reason for this disparity lies on the ratio of unlabeled anomalies in the training data. As mentioned earlier, several previous works controlled the ratio of unlabeled anomalies to be under 2%, meaning that 2% of the training data are anomalous while the rest are normal. However, in our experimental settings, this ratio matches the natural ratio of anomalies in each dataset. A common approach in existing semi-supervised AD models is to treat all unlabeled data as normal instances(e.g., DevNet, FEAWAD, PReNet), and a training set with more unlabeled anomalies will certainly bring negative impact on semi-supervised AD models. So, for datasets with low contamination rate (e.g., *Mammography*, which has 2.32% anomalies), those models tend to perform well, while for the datasets with much more anomalies (e.g., *SpamBase*, which has 39.91% anomalies), their performance will decrease. However, we argue that it is challenging to ensure the quality of unlabeled training data in real-world scenarios. As shown in the experimental results, our proposed two-stage anomaly denoising and expansion strategy, combined with a denoising-aware training objective, effectively mitigates the adverse effects of anomalies within the unlabeled set, leading to better model performance.

## 4.3 Ablation Study

We conducted an ablation study to further investigate the impact of anomaly denoising and anomalous expansion in contrastive learning, as well as the parameterized bilinear distance. First, we removed each of the first two components independently to evaluate

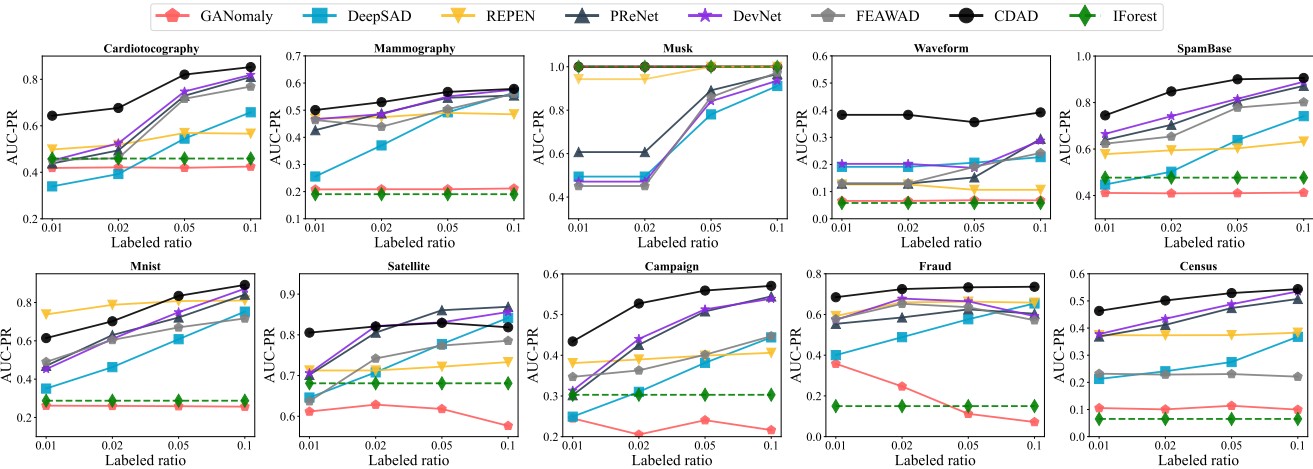

Figure 2: AUC-PR w.r.t. ratio of labeled anomalies.

their performance. The variant **w/o ND** denotes the model without normal sample denoising, and the variant **w/o ND+AE** denotes the model without the entire anomaly denoising and expansion strategy. Next, we replaced the parameterized bilinear distance with a Euclidean distance function, keeping the rest of the CAD model unchanged. This variant is denoted as **w/o PBD**. Table 3 reports the results of ablation studies.

Based on the average performance in terms of AUC-ROC and AUC-PR scores, our CAD model demonstrates the highest performance compared to three ablation variants. This underscores the effectiveness and necessity of each component. Among the three main components, we observed that removing the parameterized distance metric results in the worst performance compared to the other models. This phenomenon highlights the crucial role of the learnable distance metric in capturing complex correlations from different dimensions, which aids in identifying anomalies. Additionally, despite the absence of the parameterized distance metric, our denoising-aware contrastive distance learning framework remains competitive among baselines in Table 2., further demonstrating the effectiveness of the proposed denoising-aware training objective.

Additionally, although the anomaly denoising and expansion strategy generally enhances anomaly detection performance across most datasets, there are still a few datasets (e.g., Mammography) where this strategy slightly degrades the overall model performance. This issue may stem from the number of expanded anomalies and the accuracy of unsupervised anomaly detection (AD) models used for normal sample denoising. For datasets with a low ratio of unlabeled anomalies, an anomaly threshold of $\mu + 3\sigma$ may still misclassify some normal samples as anomalies, excluding them from training. This exclusion can hinder the model's ability to learn the relationships between normal and anomalous samples. Even so, our anomaly denoising and expansion strategy brings performance gains in the average performance across 10 datasets. We will investigate the impact of $\sigma$ in our experiments.

## 4.4 Data Efficiency

The goal of semi-supervised AD methods is to make full use of the limited labeled data. Given the difficulty of obtaining labeled data,

Table 3: Ablation study on model components. Best model are marked in bold. Cardio. refers to the Cardiotocography dataset, Mammo. refers to the Mammography dataset.

| Datasets | AUC-ROC | | | | AUC-PR | | | |
|---|---|---|---|---|---|---|---|---|
| | w/o PBD | w/o ND | w/o ND+AE | CAD | w/o PBD | w/o ND | w/o ND+AE | CAD |
| Cardio. | 0.8691 | 0.9167 | 0.9151 | **0.9264** | 0.7368 | 0.7861 | 0.7856 | **0.7929** |
| Mammo. | 0.8942 | 0.9061 | 0.9069 | **0.9092** | 0.5533 | 0.5794 | **0.5879** | 0.5618 |
| Musk | **1.0000** | **1.0000** | 0.9800 | **1.0000** | **1.0000** | **1.0000** | 0.9035 | **1.0000** |
| Waveform | 0.7614 | 0.8564 | 0.8329 | **0.8834** | 0.1493 | 0.3417 | 0.3300 | **0.3420** |
| SpamBase | 0.6993 | 0.7478 | 0.7349 | **0.7823** | 0.6291 | 0.7010 | 0.6850 | **0.7297** |
| Mnist | 0.8671 | 0.9308 | 0.9188 | **0.9676** | 0.6763 | 0.7502 | 0.7311 | **0.8247** |
| Satellite | 0.7800 | **0.8577** | 0.8508 | 0.8543 | 0.7277 | 0.7906 | 0.7741 | **0.8069** |
| Campaign | 0.6988 | 0.8588 | 0.8126 | **0.8718** | 0.2933 | 0.4515 | 0.3831 | **0.4707** |
| Fraud | 0.9561 | 0.9606 | 0.9567 | **0.9734** | 0.6087 | 0.6743 | 0.7225 | **0.7423** |
| Census | 0.7955 | **0.8955** | 0.8740 | 0.8949 | 0.3430 | **0.4554** | 0.3688 | 0.4543 |
| Average | 0.8322 | 0.8930 | 0.8783 | **0.9063** | 0.5718 | 0.6530 | 0.6272 | **0.6725** |
| P-value | 0.0018 | 0.0191 | 0.0024 | - | 0.0009 | 0.0711 | 0.0090 | - |

a superior model should achieve higher performance with fewer labeled samples. In this section, we examine the models' ability of utilizing labeled data by training the models with different labeled ratios. In particular, we vary the ratio of labeled anomalies to [0.01, 0.02, 0.05, 0.1] relative to the number of anomalies in training data to assess each model's performance. Additionally, to understand the effect of using additional information of labeled data, we include a popular unsupervised method IForest [21] for comparison.

As seen in Figure 2, semi-supervised methods generally outperform the unsupervised approach with as little as 2% labeled anomalies on most datasets. This observation aligns with the core objective of semi-supervised models: leveraging limited label information to enhance performance significantly. Notably, the performance gain when increasing the labeled data from 1% to 5% is typically more substantial than the improvement observed when moving from 5% to 10%. This trend indicates the importance of even a small amount of labeled data in semi-supervised AD. Furthermore, CAD demonstrates superior data efficiency compared to other semi-supervised approaches. Across the majority of datasets,

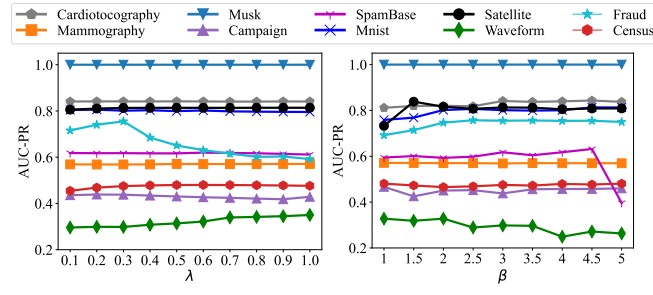

Figure 3: Effects of $\lambda$ and $\beta$.

CAD achieves better performance with fewer labeled samples. In particular, CAD utilizes only half labeled data to outperform the other models in *Cardiotocography*, *SpamBase*, *Waveform*, *Campaign* and *Fraud*. These results indicate that our framework effectively exploits the underlying data relationships even with limited labeled data, making it particularly suitable for real-world scenarios where labeled anomalies are scarce and costly to obtain.

## 4.5 Parameter Analysis

We conduct additional experiments to explore the impact of three key hyperparameters on our CAD model: the number of channels $c$ in bilinear tensor distance (Eq 6), the loss weight for expanded anomalies $\lambda$ (Eq 5), and anomaly threshold factor $\alpha$ (Eq 1). We systematically vary each hyperparameter while keeping the others at their default values and using a fixed random seed. The model's performance is recorded in terms of AUC-PR scores.

**Loss weight for expanded anomalies ($\lambda$).** The hyperparameter $\lambda$ controls the impact of expanded anomalies to model training. A larger value of $\lambda$ increases the influence of augmented anomalies on the training of model. As shown in Figure 3, the model's performance is generally stable with respect to $\lambda$ across the majority of datasets. However, a notable drop in performance is observed in the *fraud* dataset when $\lambda$ exceeds 0.3. This can be attributed to the low anomaly ratio in this dataset (0.17%), where the anomaly threshold set at $\mu + 3\sigma$ during the normal sample denoising stage still results in an overly large $k_A$. Consequently, a significant number of normal samples are mislabeled as pseudo anomalies. As $\lambda$ increases, the influence of these misclassified samples grows, leading to a decline in model performance. This observation also underscores the importance of selecting an appropriate $\alpha$.

**Number of channels in bilinear tensor distance ($c$).** The parameter $c$ control the number of channels to capture the underlying structure of the data. Given the representation size of $d_h$, we define the number of channels as $c = \beta d_h$. The right subfigure of Figure 3 illustrate the impact of varying $\beta$. As shown, the model's performance improves across several datasets when $c$ is relatively small. This suggests that a distance function with fewer channels may lack the capacity to adequately capture correlations between dimensions. However, as $c$ continues to increase, performance plateaus or even deteriorates, as seen in the *SpamBase* dataset. This decline in performance likely indicates the onset of overfitting due to the increased number of parameters. Considering the balance between model expressiveness and computational efficiency, the experimental results suggest that a $\beta$ value in the range of $[2, 4]$ provides stable and better model performance.

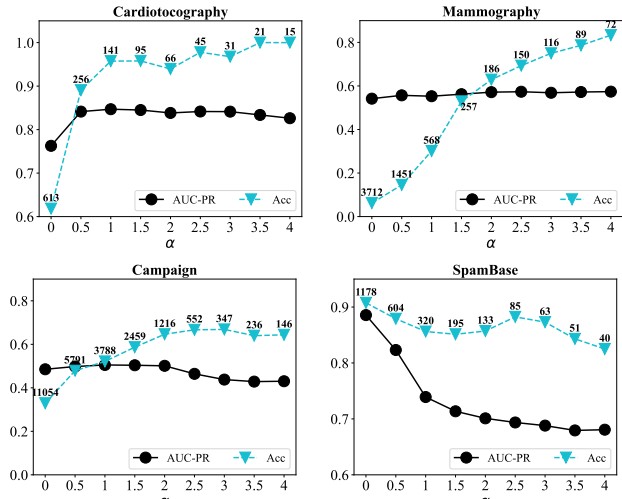

Figure 4: Effects of $\alpha$. $k_A$ is indicated on *Acc* curve.

**Anomaly threshold factor ($\alpha$).** The hyperparameter $\alpha$ controls the size of $\mathcal{X}_{\hat{U}}$ and $\mathcal{X}_{A'}$. To assess the impact of $\alpha$, we evaluate the AUC-PR, the accuracy of expanded anomalies (*Acc*), and $k_A$ in differnt datasets, where $Acc = \frac{1}{k_A} \sum_{x'_a \in \mathcal{X}_{A'}} \mathbb{1}_{[x'_a \text{ is an anomaly}]}$. Figure 4 illustrates the results for four representative datasets, which exhibites distinct responses to variations in $\alpha$. In datasets with low anomaly rates (e.g., *Mammography*), increasing $\alpha$ improves model performance. Conversely, in datasets with higher anomaly rates (e.g., *SpamBase*), a lower $\alpha$ value, resulting in a larger $k_A$, tends to enhance performance due to the inclusion of more true anomalies in the training set. In datasets like *Cardiotocography* and *Campaign*, the performance first improves with increasing $\alpha$, but then declines as $\alpha$ continues to rise. This pattern suggests an optimal trade-off point between pseudo-label accuracy and the number of pseudo-anomalies used in training. These findings highlight the dataset-specific sensitivity to $\alpha$, which is linked to the anomaly rate and the model's performance on each dataset. This variability also underscores the benefit of incorporating prior knowledge about the expected anomaly rate in the dataset to determine $k_A$.

## 5 Conclusion

In this paper, we presented CAD, a novel denoising-aware contrastive distance learning framework for semi-supervised anomaly detection. CAD leverages a contrastive learning objective to fully utilize the relationships between data points, enhancing the model's ability to learn discriminative features. The framework incorporates with a two-stage anomaly denoising and expansion strategy that allows for robust learning in the presence of noisy data. Furthermore, by introducing a parameterized bilinear tensor distance, CAD is able to capture complex feature correlations, overcoming the limitations of conventional distance measures like Euclidean distance. Experiments demonstrate that CAD not only handle noisy data more robustly but also achieve better anomaly detection with fewer labeled samples compared with existing models. Ablation study also confirms the effectiveness of key components in CAD.

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
