# OpenReview forum: "Semi-Supervised Anomaly Detection through Denoising-Aware Contrastive Distance Learning"
_ACM.org/TheWebConf/2025/Conference — WWW 2025 Poster_

### Official Review · Reviewer_EV5q · 2024-11-05

**Novelty:** 5
**Technical Quality:** 6

**Review:**

# Overview
The paper presents a novel framework for semi-supervised anomaly detection (AD) called CAD (Contrastive distance learning). It aims to address challenges associated with existing methods, particularly the handling of unlabeled anomalies and the effectiveness of distance measures in high-dimensional data. The authors propose a two-stage anomaly denoising and expansion strategy along with a parameterized bilinear tensor distance to enhance model performance.

# Strengths
* 1. Novel Approach: The introduction of a denoising-aware contrastive distance learning framework is a significant contribution to the field of anomaly detection. It effectively combines the strengths of supervised and unsupervised learning, addressing issues with data contamination in real-world applications.

* 2. Comprehensive Experiments: The authors conducted extensive experiments on 10 diverse real-world datasets, demonstrating the robustness and effectiveness of the CAD framework compared to several state-of-the-art models. The results indicate significant improvements in detection accuracy, particularly in scenarios with noisy data.
* 3. Ablation Studies: The inclusion of ablation studies provides valuable insights into the contribution of each component of the CAD framework. This methodological rigor adds credibility to the claims made regarding the framework's effectiveness.
* 4. Relevance: The problem of anomaly detection is highly relevant across various fields, including healthcare, finance, and cybersecurity. The proposed method's applicability in real-world scenarios enhances its impact.

# Weaknesses
* 1. Training efficiency: This paper could benefit from more discussion on training efficiency of the proposed methodology compared to the baselines.
* 2. Parameter Sensitivity: The model’s performance is sensitive to hyperparameters like the anomaly threshold factor $\alpha$ and loss weight $\lambda$, which could require extensive tuning across different datasets.
* 3. Assumption of Noise Levels:The denoising strategy assumes certain noise levels in unlabeled data, which might not generalize well with datasets with unknown or highly variable contamination rates. In cases of low contamination (e.g., Mammography), the strategy offers limited benefit or slight performance drops.
* 4. Limited Theoretical Justification: The paper provides an empirical demonstration of effectiveness, but there is limited theoretical discussion on why specific hyperparameters (e.g., values for \alpha, \lambda) were chosen or how they impact different types of datasets.

**Questions:**

* 1. Could you elaborate on the theoretical motivation behind specific values chosen for $\alpha$ and $\lambda$? For instance, were these values chosen based on empirical optimization alone, or are there theoretical insights that guided these selections?
* 2. In datasets with very low contamination (e.g., Mammography), it appears that the two-stage denoising strategy may not be as effective and might even slightly degrade performance. Do you have specific recommendations for adapting CAD in cases where the dataset is likely to be “cleaner” or has a low anomaly rate?
* 3. For practitioners aiming to deploy CAD in production, could you share insights into the computational efficiency or approximate training times observed in your experiments? Additionally, have you considered practical adjustments to CAD for real-time or near-real-time anomaly detection?
* 4. The two-stage denoising relies on assumptions about the noise levels in the unlabeled data. How does CAD handle cases where the noise level is unknown or dynamically changes over time? Would it be possible to integrate an adaptive mechanism for handling variable contamination levels?

**Reviewer Confidence:**

3: The reviewer is confident but not certain that the evaluation is correct

**Scope:**

4: The work is relevant to the Web and to the track, and is of broad interest to the community

---

### Official Review · Reviewer_iau8 · 2024-11-18

**Novelty:** 4
**Technical Quality:** 4

**Review:**

This paper introduces a novel framework, CAD, for semi-supervised anomaly detection that integrates denoising-aware contrastive learning with a parameterized bilinear tensor distance metric. The proposed approach addresses challenges in leveraging unlabeled datasets, particularly the contamination with anomalies and the limitations of traditional Euclidean distance metrics. CAD employs a two-stage anomaly denoising and expansion strategy to refine training data and enhance model performance. Extensive experiments on 10 real-world datasets demonstrate CAD's superiority over state-of-the-art models in both AUC-ROC and AUC-PR metrics.

**pros.**

1. The introduction of a contrastive learning framework tailored for anomaly detection is innovative, particularly the focus on distance-based metrics rather than reconstruction or classification-based methods. The parameterized bilinear tensor distance metric is a significant improvement, offering better flexibility to model complex feature relationships compared to Euclidean metrics.

2. The use of 10 real-world datasets spanning diverse domains strengthens the claims of the paper. CAD achieves consistent improvements over baselines, including substantial gains in AUC-PR for highly imbalanced datasets like SpamBase and Fraud.

3. The paper is well-structured, with clear definitions of the problem, methodology, and evaluation metrics.

**cons.**

1. While the bilinear tensor distance metric improves expressiveness, it introduces additional computational complexity. The trade-off between accuracy and efficiency is not sufficiently explored. The precomputation of global context for anomaly inference (Eq.8) is mentioned but not elaborated on in terms of scalability for large datasets.

2. The authors only compare their method with six baseline methods, many of which are relatively outdated. To ensure a fair comparison,  they should include more advanced and state-of-the-art methods in their evaluation.

3. The reliance on dataset-specific hyperparameters like $\alpha$ and $\lambda$ may hinder generalizability and require extensive tuning for new datasets.

**Questions:**

See above.

**Reviewer Confidence:**

3: The reviewer is confident but not certain that the evaluation is correct

**Scope:**

3: The work is somewhat relevant to the Web and to the track, and is of narrow interest to a sub-community

---

### Official Review · Reviewer_WYiQ · 2024-11-28

**Novelty:** 4
**Technical Quality:** 5

**Review:**

This paper presents CAD, a semi-supervised anomaly detection framework. The authors propose a contrastive learning approach that better captures relationships between anomalies and unlabeled data points. They also use a two-stage denoising strategy to handle noisy unlabeled data. In the first stage, denoting eliminates noise in the unlabeled data using distance-based methods like CBLOF; then in the second stage, expanded anomalies are generated by dynamically identifying high-confidence anomalies to enhance training diversity. They also use a parameterized bilinear tensor distance metric for measuring complex data relationships which outperforms traditional euclidean-based measures. The methods are tested on 10 real-world datasets and the results show that CAD outperforms current methods while requiring fewer labeled examples.

Pros:
- The experiments are well-designed. The ablation studies further enhance their arguments. The performance improvements across datasets are significant.
- the idea of denoising and expansion is innovative. This functions as a data cleansing for this task.

Cons:
- The model relies on a static threshold which should be manually tuned. The authors may want to attempt some adaptive approach as they have some insights about the choice of it.
- I think one time-series data could be tested here. The proposed model looks like it can be applied to time-series data which might be a more challenging and practical case for anomalies detection.

**Questions:**

CAD relies on the learned feature space. Could domain-specific feature engineering improve its performance, or do you believe your framework eliminates the need for such preprocessing?

**Reviewer Confidence:**

3: The reviewer is confident but not certain that the evaluation is correct

**Scope:**

4: The work is relevant to the Web and to the track, and is of broad interest to the community

---

### Official Review · Reviewer_YUYW · 2024-11-29

**Novelty:** 4
**Technical Quality:** 5

**Review:**

This paper focuses on Semi-Supervised Anomaly Detection (SSAD), which aims to detect anomalous samples using limited labelled data and rich unlabelled data. However, existing methods suffer from (1) artificially setting the proportion of anomalies, (2) ignoring the presence of noise in anomalies, and (3) relying on the Euclidean distance metric. To address these issues, this paper proposes a denoising-aware Contrastive distance learning framework for semi-supervised AD (CAD). Specifically, CAD includes (1) a contrastive training objective to facilitate the learning of distinctive representations by contrasting the average distance between anomalies and unlabeled samples.(2) a parameterised bilinear tensor distance layer to learn a customized distance metric. Finally, the validity of CAD was verified by a diversity of experiments.

Strengths

Novelty：This paper consider the problem of robustness of anomaly proportion in semi-supervised anomaly detection.

Technical Quality：In this paper, suitable framework are designed to effectively solve the problems in semi-supervised anomaly detection.

Experiment：The experiments are comprehensive, including main experiments, ablation experiments, hyperparameter sensitivity experiments as well as experimental visualization and analysis.

Weaknesses：
1. The methodology of this paper is similar to that of Paper [1], please describe its differences and detail the innovation of this paper.
[1] Zhou S, Zha D, Shen X, et al. Denoising-Aware Contrastive Learning for Noisy Time Series[J]. arXiv preprint arXiv:2406.04627, 2024.
2. The Anomaly threshold factor has a tendency to have different effects on different datasets, it is explained in detail in this paper (in datasets with a low anomaly rate, the higher the value, the higher the model performance; on the contrary, in datasets with a high anomaly rate, the lower the value, the higher the performance.). However, in realistic scenarios, it is often difficult to know a priori the proportion of anomaly samples, and non-adaptive can lead to poor robustness.
3. This paper does not compare the SOTA baseline.
4. This paper does not mention the training details of the experiment, such as backbone, hyperparameter settings, etc.
5. This paper focuses on semi-supervised anomaly detection, but only progress and problems in anomaly detection are presented in Introduction.
6. There are some spelling errors, i.e. CDAD in Figure 2.

**Questions:**

as above

**Reviewer Confidence:**

3: The reviewer is confident but not certain that the evaluation is correct

**Scope:**

4: The work is relevant to the Web and to the track, and is of broad interest to the community